# Assessment of Italian Population Awareness on One-Health, Zoonoses and the Mpox Vaccine: A Nationwide Cross-Sectional Study

**DOI:** 10.3390/vaccines12030258

**Published:** 2024-03-01

**Authors:** Fabrizio Bert, Giuseppina Lo Moro, Francesco Calabrese, Valentino Barattero, Alberto Peano, Giacomo Scaioli, Roberta Siliquini

**Affiliations:** 1Department of Public Health and Pediatric Sciences, University of Turin, 10126 Turin, Italy; fabrizio.bert@unito.it (F.B.); giuseppina.lomoro@unito.it (G.L.M.); valentino.barattero@edu.unito.it (V.B.); alberto.peano@unito.it (A.P.); giacomo.scaioli@unito.it (G.S.); roberta.siliquini@unito.it (R.S.); 2Hygiene and Infection Control Unit, Azienda Sanitaria Locale Torino 3, 10098 Turin, Italy; 3Collegium Medicum, University of Social Sciences, 90-113 Lodz, Poland; 4Azienda Ospedaliera Universitaria City of Health and Science of Turin, 10126 Turin, Italy

**Keywords:** One Health, zoonoses, Mpox, general population’s knowledge

## Abstract

In recent decades, the rise of zoonotic diseases has emerged as a significant human health concern, highlighting the interconnectedness of human and animal health within the framework of the “One Health” (OH) concept. This study, conducted in Italy in 2023, sought to gauge the general population’s awareness of OH and zoonotic diseases while identifying influencing factors. Additionally, it aimed to assess awareness of an Mpox virus vaccine, particularly pertinent due to the 2022 outbreak across Europe. The online cross-sectional study encompassed 1058 participants, revealing that 54.26% were unfamiliar with OH and zoonoses. Median knowledge scores were 12 points (IQR = 9–15) for zoonoses and 8 points (IQR = 6–11) for OH. Notably, factors such as age, economic situation, healthcare employment, educational level, and health literacy significantly influenced knowledge scores. Merely 26.8% of participants were aware of the existence of an Mpox vaccine, with healthcare workers, individuals engaged in animal-related work, and non-heterosexual men demonstrating higher awareness. The findings underscored a limited public understanding of zoonotic diseases and One Health, with variations observed across specific demographic groups. Given the potential impact on public health, urgent educational initiatives are warranted. Moreover, the study highlighted a low awareness of the Mpox vaccine, emphasizing the necessity for targeted awareness campaigns directed at both professionals and the general public.

## 1. Introduction

In recent decades, zoonoses have posed a significant challenge to human health, as highlighted by the COVID-19 pandemic. These diseases, along with various pre-existing global phenomena such as limited access to clean water and food in some parts of the world, global warming, increased pollution, migration patterns, and globalization, have led to the evolution of the concept of health and the associated risks, emphasizing the close interrelation between human, animal, and environmental health. This evolving paradigm is encapsulated in the concept of “One Health” [1,2].

In September 2015, the Food and Agriculture Organization (FAO), the World Health Organization (WHO), and the World Organisation for Animal Health (OIE) implemented strategies to address zoonotic diseases, foodborne illnesses, and antimicrobial resistance [3]. They emphasize the interconnectedness of the environment, wildlife, and humans [4]. Moreover, they specified that food safety, environmental pollution, antibiotic resistance, zoonotic diseases, and shared health threats among humans, animals, and the environment, like water contamination and habitat encroachment, are key research areas of One Health [1].

Focusing on zoonosis, the term refers to all diseases that are heterogeneous in terms of pathogens, transmission routes, and clinical severity and are naturally transmitted from animals to humans.

According to data reported by the Centers for Disease Control and Prevention (CDC), more than 60% of over 1700 infectious diseases affecting humans have an animal origin, such as the recent COVID-19 pandemic, but also other diseases like MERS, HIV, avian influenza, and Ebola. These diseases, after initially appearing as sporadic events limited to rural areas, have spread to become global emergencies, evolving into endemic phases. Emerging zoonoses represent one of the growing medical and economic challenges of the last 20 years, given their impact not only on public health but also on livestock trade, tourism, and wildlife conservation [5].

In Italy, as in the European Union, cases of zoonoses show a declining trend for all diseases under surveillance; however, these data must be interpreted with caution [6].

For example, one of the new international threats is the Mpox virus: already endemic in Central Africa. Since May 2022, cases and prolonged transmission chains of Mpox have been reported in countries where the disease is not endemic and without direct epidemiological links to endemic regions (travel, mammal imports). According to updated data as of October 2022, Italy is not one of the most affected European countries, but nearly 1000 cases have still been recorded [7].

Primary prevention strategies for controlling zoonoses include data collection and alarm systems on one hand and community vaccination and information campaigns on the other. Among community education programs, there is the introduction of good breeding practices and biosecurity manuals, new animal slaughter practices, and the use of personal preventive measures to avoid or reduce exposure to vector-borne diseases and other zoonoses. Depending on the disease and the availability of a vaccine, vaccination campaigns may be conducted, primarily targeting humans, livestock, or pets. Unfortunately, few vaccines are available. On one hand, this is due to the fact that the responsible microorganisms are predominantly emerging pathogens, and on the other hand, the most commonly adopted production technique (vaccines with live attenuated or inactivated viruses) hinders the development of new vaccines [8].

Therefore, the present study was designed with the primary aim of exploring the general population’s knowledge regarding One Health and Zoonoses, delving not only into their knowledge but also potential predictors and determinants, with the ultimate goal of identifying any subgroups that could hypothetically be targeted for future awareness and health education campaigns. Some authors have begun to investigate these topics; however, few have done so from the perspective of the general population, often focusing on subgroups such as students [9,10] or pet owners [11,12].

Secondarily, this study focused on assessing awareness of the existence of a vaccine against the Monkey Pox virus, a pathogen that spread significantly across various European countries, especially during the summer of 2022. The questionnaire was distributed during that period and, therefore, serves as a good proxy for evaluating the effectiveness of both official and unofficial information sources in reaching the general population.

## 2. Materials and Methods

### 2.1. Study Design

This study is a cross-sectional survey conducted through the administration of a questionnaire written in the Italian language and distributed on major social media platforms (Facebook, Twitter, WhatsApp, etc.) from 1 March 2023 to 1 May 2023. The questionnaire was set on the LimeSurvey platform, provided by the University of Turin. No information allowing the identification of the participant nor Internet Protocol (IP) addresses has been recorded. Only individuals aged 18 years or older were eligible to participate in the questionnaire. Participants completed the questionnaire voluntarily, anonymously, and without compensation. The privacy of study participants was ensured by the complete anonymity of the questionnaire, and the authors of the study were unable to trace sensitive data back to the participants. The sampling method employed was opportunistic. The questionnaire has been posted on neutral social groups, not focusing on one health, zoonoses, vaccines, or related themes to avoid selection bias.

The study was approved by the ethics committee of the University of Turin.

### 2.2. Questionnaire Structure

To collect data for the following study, a questionnaire consisting of 45 questions divided into 4 parts was constructed. Prior to questionnaire administration, information regarding the purpose, objectives, and characteristics of the study was provided, and participants were asked for their informed consent by continuing with the questionnaire.

The first part allows demographic information from patients to be obtained, such as gender, age group, occupation, education level, marital status, and information regarding the ownership of pets or livestock. To assess health literacy, the single item literacy screener (SILS) was used [13]. It consists of a single question with responses ranging from zero (Never) to four (Always). Scores higher than two indicate inadequate health literacy.

Additionally, questions were formulated to assess family size and the presence of children under five years old. The choice to evaluate this type of information was based on scientific evidence that family size and the presence of children in the household increase the risk of infection for various diseases, such as influenza, where transmission from children to adults in a family is frequent, as observed in the study by Commodari et al. [14].

In the first part of the questionnaire, anxiety levels were also assessed using the Generalized Anxiety Disorders-2 (GAD-2) scale [15]. Higher scores represent higher levels of anxiety symptoms.

The second part was dedicated to knowledge about the One Health and zoonosis topics. It consists of multiple-choice questions and a series of true or false statements. To assess knowledge in the field of One Health, we did not have questionnaires available for the general population, as they have all been conducted on highly selected populations, particularly livestock breeders. Therefore, we used some questions from Kim et al.’s survey as a reference [16], and for questions related to antibiotic resistance, we used some from the WHO’s 2015 survey on the general population’s awareness of this topic [17]. The total number of questions for the One Health theme is 15, while for the zoonosis theme, it amounts to 19. The scores on knowledge about One Health and zoonoses represented the main outcomes of this paper. Furthermore, sources of information concerning these topics were explored.

The third part is dedicated to vaccines. Participants were asked if they were aware of the existence of a vaccine against Mpox. Being aware of the existence of this vaccine was our last outcome.

The questionnaire also included information regarding participants’ level of concern about the possibility of contracting a zoonotic disease, attitudes toward several vaccinations, and adherence to conspiracy theories.

This last part of the questionnaire will be described in another article and was not evaluated in the present paper, which instead focuses on knowledge of One Health and zoonosis topics.

The part of the questionnaire used for the present paper is available in the Appendix A (both Italian and English versions).

### 2.3. Statistical Methodology

For the scores on zoonosis and One Health, we calculated the number of correct answers for each participant; unanswered or incorrect responses did not reduce the score achieved by each participant.

The Cronbach’s alpha coefficient was calculated to assess the consistency of the questions in understanding the level of knowledge on zoonoses and One Health.

For continuous variables, we computed the mean, median, first and third quartiles. For outcome variables (One Health and zoonosis scores), we used the Shapiro–Wilk test to assess the normality of the distribution and decide whether to use parametric or non-parametric tests. Furthermore, for each continuous primary outcome, we assessed the median and interquartile range (IQR), stratifying by each categorical independent variable.

Considering the non-normal distribution of the outcomes, we employed the following:-Wilcoxon–Mann–Whitney test for unpaired samples to evaluate the effect of categorical and dichotomous independent variables.-Analysis of variance with the Kruskal–Wallis test to assess the effect of non-dichotomous categorical independent variables.-Spearman rank correlation to evaluate the correlation between age (independent variable) and scores.

Statistical tests were conducted using a two-sided approach, with a significance level set at 95%.

Moreover, multiple logistic regression analyses were conducted to assess the impact of population characteristics on knowledge of zoonoses and One Health. The choice of the logistic model is related to the particular type of dependent variable we measured. The scores have two characteristics that need to be considered:-Scores cannot assume a value less than 0, because they are the result of summing the correct answers;-Scores have an upper limit.

For this reason, we chose a logistic model (Equation (Equation 1)), reporting the results as coefficients accompanied by the 95% confidence interval:(1)$$log\frac{Correct\;Responses}{Incorrect\;Responses}=\beta_{0}+\beta_{1}x_{1}+\cdots +\beta_{n}x_{n}$$

For the analysis of participants’ knowledge of the existence of a vaccine against Mpox, a chi-square test was used. Multiple logistic regression analyses were conducted to assess the impact of various covariates on knowledge of the Mpox vaccine, expressing the results in terms of Odds Ratios accompanied by the 95% confidence interval (CI). Regression models were compared using the Likelihood Ratio Test (LRT), a statistical test used to assess if the inclusion of additional parameters in a more complex model significantly improves its fit compared to a simpler model. Moreover, the Variance Inflation Factor (VIF) was calculated for each model to exclude multicollinearity. For data analysis and statistical analysis, R statistical software (version 4.2) was used [18].

## 3. Results

### 3.1. Demographic Characteristics of the Sample

The total number of participants who completed the questionnaire was 1058. The median age of the interviewed population was 33 years, ranging from 25 to 46. Of the total population, 51.7% were female, while 47.3% were male.

Regarding sexual orientation, 89.5% identified as heterosexual, 3.7% as homosexual, 5.2% as bisexual, and the remaining 1.6% identify as other or asexual.

The majority of respondents (80.7%) stated that they were either employed or students, with 16.7% working in healthcare, and 5.9% of them working with animals. A total of 79.5% of the respondents have owned or currently own pets.

The SILS score suggested that 74.2% of the sample had adequate health literacy, while the average level of anxiety measured using the GAD-2 score was 2.32.

### 3.2. Information Sources for One Health and Zoonoses

Analyzing the information sources through which participants received information about zoonoses or One Health, it emerges that 54.26% had never heard of these topics (Table 1). The remaining 45.74% had heard about them primarily in an educational environment (7.44%), from friends (6.80%), through the literature (6.37%), or on social media (10.38%). General practitioner and veterinarians were less prominent as sources of information.

As observed in Table 2, the majority of participants indicated social media (12.67%) and their general practitioner (12.11%) as their preferred sources to potentially receive information when it comes to zoonoses and One Health topics.

Television programs, websites, and videos were also considered as sources from which they would like to receive information.

In contrast, just over 3% of participants would prefer to receive information during working hours, while 6.24% would not prefer to receive any type of information on these topics.

### 3.3. Zoonosis Scores

The median score obtained by participants was 12 out of 19 points (63%) with an interquartile range between 9 and 15 points. The consistency of the score on zoonoses, obtained through the Cronbach’s coefficient, was 0.81.

Considering individual responses regarding the topic of zoonoses, only 373 participants, representing 35.2% of the total, identified Ebola as a zoonosis, and 294 participants, comprising 27.8% of the total, were aware of the prevalence of zoonoses compared to the total number of infectious diseases. A slightly higher percentage, 43.2%, equivalent to 457 out of 1058 participants, believes that some zoonoses are preventable through vaccination.

When relating sociodemographic characteristics to outcomes (Table 3), it can be observed that the distribution was not significantly different by gender (p=0.827), sexual orientation (p=0.825), and the absence of vulnerable individuals within the family nucleus (p=0.460). Additionally, the population of the town of residence and marital status influenced knowledge on the subject of zoonoses.

On the other hand, Italian nationality, employment status, marital status, economic situation, population of the place of residence, and health literacy were characteristics that influenced knowledge on this subject.

Finally, assessing some specific characteristics related to work and cultural background, we observed a difference in the distribution of zoonosis scores. In particular, those working in the healthcare sector tend to score 4 points higher (p≤0.001), as do graduates compared to non-graduates (14 versus 12, p≤0.001), and those working with animals (p≤0.001) (Table 3). A negative correlation was also observed (−0.14, p≤0.001) between age and the total score.

### 3.4. One-Health Scores

The median total score was 8 out of 15 points, equivalent to 53%, with an interquartile range between 6 and 11 points. The Cronbach’s coefficient yielded a consistency score of 0.74 for the One-Health domain. As can be seen in Table 3, considering gender and sexual orientation, there is no difference in the score distribution. Geographical origin, both in terms of nationality and region of origin, had an impact on knowledge of the One-Health topic.

In addition, when evaluating individual responses on the topic of One Health, only 250 participants, representing 23.6% of the total, were aware that human activity impacts the emergence of new diseases, while fewer than 350 participants, representing about 30% of the total, were aware that One Health also addresses the psychological well-being of workers and secondary prevention.

In particular, in our study, we observed a higher median score in those with Italian nationality compared to those from a different country (13 points versus 10 points), and we also observed a lower median score in those residing in northern Italy (12 points versus 10 points). When considering factors such as population density of the place of residence, the presence or absence of vulnerable individuals in the family, marital status, and pet ownership, no differences in scores on the One-Health topic were observed.

On the contrary, various sociodemographic characteristics considered significantly influence knowledge on the One-Health topic. In particular, working in the healthcare sector leads to an almost 30% increase in the average score (p≤0.001) and having a degree increases the average score by 2 points (p≤0.001), as does working with animals (p≤0.001). A high level of health literacy also results in an average score increase of 2 points (p≤0.001), as does a good or excellent economic situation (p≤0.001).

Finally, a negative correlation was observed (−0.14, p≤0.001) between age and the One-Health score.

### 3.5. Multiple Logistic Regression: Zoonosis Score

As can be seen in Table 4, an initial analysis (Model 1) was conducted to understand how demographic factors (age, gender, presence of children, economic situation, population density of the town, and presence of vulnerable family members) influence the knowledge of the general population about zoonoses. In this context, the coefficient β for age was negative, −0.012 (95% CI: −0.014, −0.009), as was the coefficient β for the absence of vulnerable family members, −0.140 (95% CI: −0.200, −0.080). Conversely, a positive β coefficient could be observed for variables such as good or excellent economic situation, 0.402 (95% CI: 0.327, 0.476), and a population density of the town of more than 50,000 inhabitants, 0.285 (95% CI: 0.225, 0.345).

Furthermore, a model was created (Model 2) that considered only cultural and educational variables, such as being a healthcare worker and the level of health literacy. This model highlighted that working with animals and being healthcare workers significantly increased knowledge about zoonoses (p≤0.001), as does having obtained a degree. Conversely, the β coefficient related to low health literacy was negative, −0.229 (95% CI: −0.296, −0.161).

Finally, in Model 3 of Table 4, all the covariates used in the previous models were included. Although there were slight variations in the β coefficients, both in sign and statistical significance, the main patterns remained unchanged, except for the β coefficient related to male gender, which becomes positive, 0.099 (95% CI: 0.039, 0.159).

When comparing the “reduced” models to Model 3 using a likelihood ratio test (LRT), the deviance of the residuals was lower in the latter, both compared to Model 1 (p≤0.001) and Model 2 (p≤0.001). This indicates that Model 3 captures a greater proportion of the data variability and offers a more comprehensive explanation for the observed outcomes compared to both Model 1 and Model 2.

Attempts to introduce pet ownership into the models did not improve the deviance of the residuals, so it was decided not to include this variable in the proposed models.

### 3.6. Multiple Logistic Regression for One Health Knowledge Score

Similar to the analysis conducted for the zoonosis score, several regression analyses were performed (Table 5). A preliminary analysis (Model 1) aimed to understand how demographic factors (age, gender, presence of children or fragile individuals, economic situation, and population size of the municipality) influence the general population’s knowledge of One Health.

All examined factors were associated with knowledge of One Health, except for the presence of children. Specifically, a good economic situation (β=0.586, 95% CI: 0.502, 0.670) and the population size of the municipality (β=0.068, 95% CI: 0.002, 0.133) were associated with greater knowledge on this topic. Conversely, factors associated with lower knowledge of zoonosis were the absence of fragile individuals in the household (p≤0.005), a different gender compared to female, and age (p≤0.001).

Additionally, a model (Model 2) was created that considered only cultural and educational variables, such as education level and working with animals. All covariates considered significantly influenced knowledge of One Health. In particular, being a healthcare worker (β=0.747, 95% CI: 0.652, 0.841), working with animals (β=0.150, 95% CI: 0.009, 0.290), and having a bachelor’s or higher degree (β=0.265, 95% CI: 0.192, 0.338) were associated with greater knowledge of One Health. On the other hand, being unemployed and having low health literacy were negatively correlated with knowledge on this topic.

Finally, in Model 3 of Table 4, all the covariates used in the previous models were included. In this case, changes in the significance of some coefficients were observed. In particular, being male no longer correlated with the level of knowledge, nor did the population size of the municipality of residence. Even working with animals and the absence of employment no longer showed any correlation with knowledge of One Health. However, the presence of children remained significantly correlated (β=0.097, 95% CI: 0.004, 0.191).

Comparing the first two models with Model 3 through a likelihood ratio test (LRT), the deviance of residuals was lower in the latter, both compared to Model 1 (p≤0.001) and Model 2 (p≤0.001). This leads to the same considerations that have already been made in the previous paragraph about the comparison between models regarding knowledge about zoonoses.

### 3.7. Mpox Vaccine Knowledge

The prevalence of participants aware of the existence of a vaccine for Mpox was 26.8% (283 out of 1055).

As can be seen in Table 6, for some socio-demographic variables, there was no difference in knowledge of the existence of a vaccine for Mpox virus. In particular, gender, sexual orientation, and nationality did not seem to influence whether participants knew that there was a vaccine for Mpox. A similar observation can be made for factors such as occupation, having children or vulnerable individuals in the family, marital status, and pet ownership. Even a good level of health literacy did not appear to be associated with an increase in the prevalence of individuals aware of the availability of this type of vaccine.

On the contrary, being healthcare professionals or working closely with animals increased the prevalence of those who know that there was a vaccine for the Mpox virus (p<0.001), as well as having a higher level of education (p<0.001), regional origin (p=0.012), a good economic situation (p=0.002), and living in a municipality with more than 50,000 inhabitants (p=0.038).

Several logistic regression analyses were conducted using the knowledge of the existence of the Mpox virus vaccine as the dependent variable to assess the impact of various covariates. In particular, three models were created to further investigate the effect of certain variables that might influence the dependent variable, such as age, gender, and sexual orientation.

The first model considered these three variables independently, without interactions. As shown in Table 7, none of the examined covariates had a significant impact on the dependent variable.

In contrast, in the second model (Table 7), a relationship was introduced between gender and sexual orientation. The result was that these two covariates influenced each other, and although the statistical significance remained unchanged for the individual covariates, the model revealed that non-heterosexual males were more aware of the existence of a vaccine for the Mpox virus, with an odds ratio of 2.788 (1.091, 7.283).

The addition of new covariates did not change this relationship, as can be seen in Model 3, which exhibits a significantly lower deviance of residuals (p<0.001). Furthermore, the third model revealed a significant impact of education level (OR=1.643; 95% CI: 1.215, 2.218), being a healthcare worker (OR=2.386; 95% CI: 1.663, 3.419), and working with animals (OR=2.291; 95% CI: 1.328, 3.927).

## 4. Discussion

### 4.1. Knowledge about Zoonoses and One Health

Our study has highlighted that the majority of participants had never heard of zoonoses or One Health. This was reflected in the scores we measured. In fact, for most of the questions, participants could respond with true or false, or they could indicate that they did not know the answer. This explains the abundance of scores equal to 0 points. Considering the overall scores, participants on average answered correctly to 8 out of 15 questions for the One Health score and 12 out of 19 questions for the zoonoses score. This indicates a lack of basic information on the topic within the population. Similar results had also emerged from studies on other populations, such as the one by Kim et al. [16], based on the rural population of the Philippines, where only a minority of interviewed families (2.2%) indicated that they had heard the term “One Health”, even though many had basic knowledge of fundamental OH concepts related to health promotion in the human–animal–environment interface.

### 4.2. Information about Zoonoses and One Health

Our study reveals that the majority of individuals had never heard of these topics. The sources through which the population was currently informed and updated on these issues were diverse and varied.

It was interesting to note that few participants receive information from their general practitioner or veterinarian. On the contrary, the media (news broadcasts, social media) collectively represent the major source of information.

Focusing on participant preferences, it becomes clear that many favor social media as their primary means of receiving information. This was undoubtedly linked to the average age of our sample, which does not represent the average age of the Italian or European population.

Nevertheless, many still prefer to receive information from their general practitioner or through the use of television programs, videos, or websites.

Therefore, in promoting informational campaigns, it was essential in this context for health authorities to utilize digital as well as diverse information channels, ranging from social media to television programs.

It was important in this context to emphasize that general practitioners and veterinarians should also have a greater role in promoting information on these topics.

### 4.3. Determinants of Zoonosis Awareness

Analyzing the results, younger participants appear to have more information about the topic of zoonoses. As shown in Table 3, Italian nationality seems to be a factor influencing the number of correct answers, but this was likely related to a better understanding of the language, given the complexity of the topic.

As expected, healthcare professionals, those working with animals, and individuals with higher levels of education demonstrate greater knowledge of the subject. A good level of health literacy also generally results in higher scores.

Interestingly, being a pet owner does not play a role in knowledge of diseases that can be transmitted by animals, contrary to what Pereira et al. [19] had highlighted. Some demographic variables analyzed seem to influence knowledge of zoonoses, but establishing a causal relationship was challenging. It was difficult to understand how the size of the municipality or economic status can influence awareness of this topic, as well as the presence of children in the family. It can be hypothesized that a higher socio-economic status provides more resources and opportunities to stay informed about such topics.

It was also possible that there were interactions between demographic variables that influence these results, which was why we had decided to propose various multiple analyses.

In particular, three different models were presented (Table 4) that explore the impact of demographic and socio-cultural variables. It can be appreciated that Model 3 performs better in terms of predictiveness compared to the reduced models. Furthermore, we do not observe significant changes in the β coefficients, allowing us to rule out gross relationships between the model covariates.

Considering Model 3, it emerges that the knowledge of zoonoses decreases with age, as well as in individuals with low health literacy and those without vulnerable family members. On the contrary, good economic status, as well as a high level of education, working with animals, or a career in healthcare, improves knowledge of zoonoses. These results were very similar to those reported in other studies investigating knowledge of zoonoses in specific subject categories [20,21].

The male gender variable also slightly improves the score obtained, although it was difficult to explain since we had no reason to believe that gender can influence knowledge of the topic.

Furthermore, the unemployed status, which appears to improve knowledge of zoonoses, was difficult to explain, but given the participants’ average age, many may still be in university education.

Finally, the absence of vulnerable family members correlates negatively, while the presence of children in the family was positively correlated with knowledge of zoonoses. While the absence of awareness linked to lower interest in the topic may be hypothesized for the first variable, it was difficult to find an explanation for the second variable considering that the model already takes age into account.

In conclusion, older individuals with a less favorable economic situation and lower levels of education had less information about zoonoses. It was interesting to note that owning one or more pets does not change the level of knowledge.

### 4.4. Determinants of One Health Knowledge

As observed in Table 3, various demographic and socio-cultural variables seem to influence the scores that participants achieve on the topic of OH.

Gender and sexual orientation do not appear to play a role in knowledge of the topic, and it was interesting to note that, for example, the presence of vulnerable family members has no impact on the score, even though some of these individuals work as caregivers.

On the contrary, as expected, working with animals, being a healthcare professional, and having good health literacy increase the number of correct answers provided by participants. Higher education levels also appear to influence knowledge on the topic, likely by reducing the possibility of assimilating incorrect information, in line with what was reported by the Italian communications authority in a 2020 report [22].

Although there was evidence regarding this [11], it was difficult to understand whether employment status and economic situation had a direct role in knowledge of One Health, or if the relationship highlighted was linked to a third independent variable that influences both the socio-demographic variable and knowledge about OH.

For some variables, the relationship with knowledge of One Health was difficult to interpret, such as the region of residence. This was why several regression models were proposed. In this context, it was useful to comment on Model 3 from Table 5.

In particular, it emerges that with an increase in the participant’s age, there was a reduction in knowledge on this topic, as well as the absence of vulnerable individuals in the family and low health literacy. Significantly correlated with greater knowledge of One Health were higher education levels, the presence of children, and being a healthcare professional.

In the context of this model, employment status and working with animals lose statistical significance. On the other hand, a good or excellent economic situation remains positively correlated with knowledge of One Health.

Considering the coefficients of Model 3, it was possible to make a prediction about knowledge of the One Health topic. In particular, two indicative examples can be proposed:

A 30-year-old healthcare professional with a degree, in a good economic situation, living in a small municipality, with no children or vulnerable individuals in the family, employed, and with a good level of health literacy, should, according to the model, achieve a score of 11.3 out of 15. In contrast, a demographically similar individual, but without a degree, not working in healthcare, and with low health literacy, according to the model, would achieve a score of 7.3 out of 15. All of this suggests that older people, those not working in healthcare, without a degree, and with a less favorable economic situation could have less knowledge of the One Health topic. Finally, it was interesting to note that, contrary to what was observed for knowledge of zoonoses, the One Health topic was not well known among those who work with animals.

### 4.5. Determinants of Monkey Pox Vaccine Knowledge

The second part of the study focuses on the awareness of the general population regarding the existence of a vaccine for Monkey Pox. Specifically, the questionnaire was distributed starting from the spring of 2022, a period characterized by an outbreak of Mpox cases [23] in several European states.

This phenomenon received significant media attention, and it was interesting to examine whether, the general population was aware of the existence of a vaccine against Monkey Pox.

What emerges from our study was that the majority of participants (73.2%) were not aware of the existence of a vaccine for this pathogen. Additionally, some demographic or social variables should be correlated with greater awareness.

In particular, our study shows that both healthcare professionals and those working closely with animals were more aware of the existence of a vaccine for Monkey Pox. However, it was interesting to note that less than 45% of healthcare professionals were aware of the vaccine’s existence.

Similar to knowledge of One Health and zoonosis topics, in the context of Monkey Pox vaccination, a good economic status and a high level of education were associated with greater awareness. Additionally, in this case, it can be hypothesized that those living in medium to large-sized municipalities were more easily informed on this topic.

Again, as in the previous sections, it was difficult to understand why in northern Italy, the prevalence of participants who were aware of the vaccine’s existence was higher.

Finally, considering that the ECDC defined the transmission of Mpox in the summer of 2022 as more likely among young to middle-aged men aged 18 to 50 who had sex with other men [23], one might expect greater awareness of the vaccine’s existence in the homosexual population. However, this does not emerge from the univariate analysis.

For this reason, in Models 2 and 3 (Table 7), an interaction between the gender variable and the variable identifying sexual orientation was hypothesized. Indeed, observing, for example, the third model, it can be seen that the odds ratio for vaccine knowledge was not only significant but was more relevant for non-heterosexual males compared to the ORs related to being a healthcare professional or working with animals. Considering representativeness, it should be noted that, in our sample, the percentage of individuals who self-identified as non-heterosexual was just slightly higher compared to the latest available data, which indicate that the LGBT+ population accounts for 9% in Italy [24].

Furthermore, the model confirms what essentially emerged in the univariate analyses, namely that a high level of education increases the likelihood of knowing about the vaccine by 60%, while working in healthcare and working in direct contact with animals double this likelihood.

What can be deduced from the above was that the existence of the vaccine against Monkey Pox was not known by the majority of participants, including 55% of healthcare workers and 52% of those working with animals. This findings are analogous to those made by another study conducted in Italy on healthcare workers [25]. Moreover, these results do not differ consistently from what has been reported in other studies conducted in other European countries such as the Netherlands and Denmark [26,27], where it has been shown that within specific population groups, particularly in MSM, the awarness toward Monkey Pox and willingness to vaccinate turns out to be significant.

Nevertheless, as highlighted in studies conducted in a socio-cultural context distinct from that of Europe [28,29], there is a clear need to encourage additional training initiatives in this field. This applies to both professionals and the general population, particularly focusing on individuals at risk.

### 4.6. Strengths and Limitations of the Study

This study had several limitations. First, it was conducted through the diffusion of an online questionnaire via social networks. This represents a limitation because only those with internet access and membership in one of the social networks described in the methods could complete the questionnaire. Therefore, it is difficult to generalize the results to the entire population. Also, it should be noted that through this opportunistic sampling method, we were unable to determine the total number of people reached by the questionnaire.

Furthermore, the structure of the questionnaire, especially the part aimed at measuring knowledge about zoonoses and One Health, was to a large extent of a true-false type. Thus, while participants could claim not to know the answer, it is also possible to answer correctly to half of the questions by responding entirely randomly. This reduces the questionnaire’s ability to distinguish those who genuinely understand the topic from those who do not. In addition, it should be noted that, although these outcomes’ scores had good reliability, they were not validated.

Last, it is challenging to assess why certain demographic variables significantly impact the scores and the potential presence of recall bias cannot be excluded.

However, this study is one of the first studies that investigate the knowledge of phenomena such as zoonoses or topics like One Health among the general population and it had a good sample size.

Moreover, this study identifies demographic, social, and economic variables that influence knowledge of these topics and suggested the identification of one or more target populations for informational initiatives. In addition, few studies investigated knowledge of the Monkey Pox vaccine and explored the demographic variables that influence this knowledge.

## 5. Conclusions

The present study found that understanding of zoonoses and One Health was limited in the general population, with significant differences based on demographic and economic characteristics.

Since it has been revealed that the lack of a culture regarding issues such as antibiotic resistance, the gradual depletion of environmental resources, food safety, climate change, and zoonoses can have a significant impact on public health. It was essential, therefore, to act in terms of education in this regard [26,30,31].

Last, our study uncovers a concerning lack of awareness among the general population regarding the Monkey Pox vaccine, even in light of the media attention during the 2022 outbreak. Although certain groups, such as healthcare professionals and those engaged in animal-related work, exhibit higher levels of awareness, the overall understanding remains insufficient, even among healthcare workers themselves. This underscores the pressing need for targeted educational campaigns aimed at both professionals and the broader public, especially those at risk.

## Figures and Tables

**Table 1 vaccines-12-00258-t001:** Information sources for One Health and zoonoses.

Information Source	Count ^1^	Percentage
Never heard of	758	54.26%
School	104	7.44%
Friends	95	6.80%
Literature	89	6.37%
Social Media (official pages)	77	5.51%
Social Media	68	4.87%
TV News	45	3.22%
Newspapers	40	2.86%
Veterinarian	37	2.65%
Entertainment	32	2.29%
Ministerial Website	22	1.57%
General Practitioner	19	1.36%
Pharmacist	8	0.57%
Radio	3	0.21%
**Total**	**1397**	**100%**

^1^ Each participant could provide more than one response.

**Table 2 vaccines-12-00258-t002:** Preference of information sources for One Health and zoonoses.

Information Source Preference	Count ^1^	Percentage
Social Media	337	12.67%
General Practitioner	322	12.11%
Television Programs	303	11.39%
Videos	283	10.64%
Websites	274	10.30%
Leaflets	228	8.57%
Infographics	196	7.37%
Advertisements	181	6.80%
Events organized by local health authority	172	6.47%
Never	166	6.24%
Veterinarian	112	4.21%
During working hours	86	3.23%
**Total**	**2660**	**100%**

^1^ Each participant could provide more than one response.

**Table 3 vaccines-12-00258-t003:** Influence of socioeconomic variables on knowledge of zoonoses and one health.

Characteristic	Sample Composition (%)	Zoonosis Score		One Health Score	
		Median (IQR)	*p*-Value	Median (IQR)	*p*-Value
**Gender**					
Male	500 (47.3)	12 (9–15)	0.827	8 (6–10)	0.627
Female	547 (51.7)	13 (9–15)		8 (6–11)	
Other	11 (1.0)	13 (7.5–16.5)		7 (4.5–8)	
**Sexual Orientation**					
Heterosexual	947 (89.5)	12 (9–15)	0.132	8 (6–11)	0.825
Other	111 (10.5)	13 (9–16)		8 (6–11)	
**Nationality**					
Italian	1006 (95.1)	13 (9–15)	**<0.001**	8 (6–11)	**<0.001**
Other	52 (4.9)	10 (6.75–13)		6 (4–9)	
**Region**					
North	599 (57)	12 (9–15)	0.148	9 (6–11)	**<0.001**
Central	229 (21.8)	13 (9–15)		8 (5–10)	
South	244 (23.2)	13 (10–15)		8 (6–10.75)	
**Education**					
Diploma or lower	748 (70.7)	12 (9–14)	**<0.001**	8 (6–10)	**<0.001**
Degree or higher	310 (29.3)	14 (11–16)		10 (7–12)	
**Employment**					
Employed	871 (82.3)	13 (9–15)	**0.015**	9 (6–11)	**<0.001**
Unemployed	187 (17.7)	12 (9–14)		7 (5–9)	
**Healthcare**					
No	881 (83.3)	12 (9–14)	**<0.001**	8 (6–10)	**<0.001**
Yes	177 (16.7)	16 (14–18)		12 (9–13)	
**Work with Animals**					
No	995 (94)	12 (9–15)	**<0.001**	8 (6–11)	**<0.001**
Yes	63 (6.0)	16 (13.5–17)		10 (7–12)	
**Children**					
No	671 (63.4)	13 (9–15)	**0.001**	9 (6–11)	**0.034**
Yes	387 (36.6)	12 (9–14)		8 (6–10)	
**Presence of Vulnerable Family Members**					
Yes	425 (40.2)	13 (10–15)	0.074	8 (6–11)	0.460
No	633 (59.8)	12 (9–15)		8 (6–11)	
**Health Literacy**					
High	785 (74.3)	13 (10–15)	**<0.001**	9 (7–11)	**<0.001**
Low	273 (25.8)	11 (8–13)		7 (5–9)	
**Population Density**					
More than 50,000	410 (38.8)	13 (10–15)	**<0.001**	9 (6–11)	0.3872
Less than 50,000	648 (61.2)	12 (9–15)		8 (6–11)	
**Marital Status**					
No relationship	404 (38.2)	12 (9–15)	**0.004**	8 (6–11)	0.6183
Relationship	654 (61.8)	13 (10–15)		8 (6–11)	
**Economic Situation**					
Good/Excellent	875 (82.7)	13 (10–15)	**<0.001**	9 (7–11)	**<0.001**
Poor/Insufficient	183 (17.3)	11 (8–13)		6 (4–9)	
**Owns Pets**					
No	217 (20.5)	13 (9–15)	0.347	8 (5–11)	0.052
Yes	841 (79.5)	12 (9–15)		8 (6–11)	

**Table 4 vaccines-12-00258-t004:** Regression models for knowledge on zoonoses.

	*Dependent Variable:*		
	log(Odds of Correct Responses/Incorrect Responses)		
	Model 1	Model 2	Model 3
	Coefficients:	Coefficients:	Coefficients:
Age	−0.012 *** (−0.014, −0.009)		−0.008 *** (−0.012, −0.005)
Male	0.001 (−0.057, 0.059)		0.099 *** (0.039, 0.159)
Non-Binary/Other	−0.270 * (−0.553, 0.013)		−0.099 (−0.387, 0.189)
Economic Situation: Good/Excellent	0.402 *** (0.327, 0.476)		0.255 *** (0.178, 0.332)
Town > 50,000 inhabitants	0.285 *** (0.225, 0.345)		0.234 *** (0.172, 0.295)
Children: Yes	0.024 (−0.059, 0.107)		0.088 ** (0.004, 0.172)
Fragile Presence: No	−0.140 *** (−0.200, −0.080)		−0.167 *** (−0.228, −0.105)
Healthcare Worker: Yes		0.910 *** (0.818, 1.003)	0.897 *** (0.802, 0.991)
Working with Animals		0.585 *** (0.443, 0.726)	0.531 *** (0.389, 0.674)
Degree or Higher		0.317 *** (0.249, 0.384)	0.278 *** (0.209, 0.347)
Unemployed		0.107 *** (0.029, 0.184)	0.170 *** (0.086, 0.254)
Low Health Literacy		−0.229 *** (−0.296, −0.161)	−0.197 *** (−0.266, −0.127)
Constant	0.571 *** (0.454, 0.689)	0.295 *** (0.250, 0.341)	0.326 *** (0.204, 0.447)
Observations	1055	1055	1055
Log Likelihood	−3721.998	−3474.426	−3394.345
Akaike Inf. Crit.	7459.995	6960.852	6814.690

Note: * *p* < 0.1; ** *p* < 0.05; *** *p* < 0.01.

**Table 5 vaccines-12-00258-t005:** Regression models for knowledge on One Health.

	*Dependent Variable:*		
	log(Odds Correct Answers/Incorrect Answers)		
	Model 1	Model 2	Model 3
	Coefficients:	Coefficients:	Coefficients:
Age	−0.011 *** (−0.014, −0.008)		−0.006 *** (−0.009, −0.002)
Men	−0.089 *** (−0.153, −0.025)		−0.020 (−0.086, 0.045)
Non-Binary/Other	−0.650 *** (−0.964, −0.335)		−0.459 *** (−0.778, −0.140)
Economic Situation: Good/Excellent	0.586 *** (0.502, 0.670)		0.456 *** (0.369, 0.542)
Municipality > 50,000 inhabitants	0.068 ** (0.002, 0.133)		0.029 (−0.038, 0.095)
Children: Yes	0.033 (−0.058, 0.125)		0.097 ** (0.004, 0.191)
Fragile Presence: No	−0.073 ** (−0.139, −0.007)		−0.123 *** (−0.191, −0.055)
Healthcare Worker: Yes		0.747 *** (0.652, 0.841)	0.729 *** (0.633, 0.826)
Working with Animals		0.150 ** (0.009, 0.290)	0.075 (−0.066, 0.216)
Bachelor’s Degree or Higher		0.265 *** (0.192, 0.338)	0.215 *** (0.140, 0.289)
Not Employed		−0.119 *** (−0.204, −0.034)	−0.077 (−0.170, 0.016)
Low Health Literacy		−0.251 *** (−0.326, −0.176)	−0.221 *** (−0.298, −0.143)
Constant	0.180 *** (0.050, 0.310)	0.090 *** (0.040, 0.140)	−0.029 (−0.163, 0.105)
Observations	1055	1055	1055
Log Likelihood	−3036.719	−2919.729	−2848.524
Akaike Information Criterion	6089.437	5851.459	5723.049

Note: ** *p* < 0.05; *** *p* < 0.01.

**Table 6 vaccines-12-00258-t006:** Socio-demographic factors in relation to Mpox vaccine knowledge.

Characteristic	Aware of Mpox Vaccine Existence	Unaware of Mpox Vaccine Existence	*p*-Value
**Gender**			
Male	137 (27.5%)	361 (72.5%)	0.890
Female	143 (26.2%)	403 (73.8%)	
Other	3 (27.3%)	8 (72.7%)	
**Sexual Orientation**			
Heterosexual	251 (26.59%)	(73.41%)	0.696
Other	(28.83%)	(71.17%)	
**Nationality**			
Italian	273 (27.22%)	730 (72.78%)	0.268
Other	10 (19.23%)	42 (80.77%)	
**Region**			
North	138 (23.35%)	453 (76.65%)	**0.012**
Central	66 (29.73%)	156 (70.27%)	
South	77 (32.77%)	158 (67.23%)	
**Education**			
Diploma or lower	174 (23.36%)	571 (76.64%)	**<0.001**
Bachelor’s degree or higher	109 (35.16%)	201 (64.84%)	
**Occupation**			
Employed	233 (26.84%)	635 (73.16%)	1
Unemployed	50 (26.74%)	137 (73.26%)	
**Healthcare Profession**			
No	204 (23.23%)	674 (76.77%)	**<0.001**
Yes	79 (44.63%)	98 (55.37%)	
**Work with Animals**			
No	253 (25.5%)	739 (74.5%)	**<0.001**
Yes	30 (47.62%)	33 (52.38%)	
**Children**			
No	182 (27.2%)	487 (72.8%)	0.768
Yes	101 (26.17%)	285 (73.83%)	
**Presence of Vulnerable Individuals in the Family**			
Yes	123 (29.01%)	301 (70.99%)	0.214
No	160 (25.36%)	471 (74.64%)	
**Health Literacy (SILS)**			
High	210 (26.79%)	574 (73.21%)	1
Low	73 (26.94%)	198 (73.06%)	
**Municipal Population**			
More than 50,000	125 (30.49%)	285 (69.51%)	**0.038**
Less than 50,000	158 (24.5%)	487 (75.5%)	
**Marital Status**			
No Relationship	98 (24.38%)	304 (75.62%)	0.181
Relationship	185 (28.33%)	468 (71.67%)	
**Economic Situation**			
Good/Excellent	251 (28.75%)	622 (71.25%)	**0.002**
Poor/Insufficient	32 (17.58%)	150 (82.42%)	
**Pet Ownership**			
Yes	223 (26.61%)	615 (73.39%)	0.824
No	60 (27.65%)	157 (72.35%)	

**Table 7 vaccines-12-00258-t007:** Regression models for Mpox vaccine awareness.

	*Dependent Variable:*		
	Awareness of Mpx Vaccine		
	Model 1	Model 2	Model 3
	(OR)	(OR)	(OR)
Age	0.994 (0.984, 1.004)	0.994 (0.984, 1.004)	1.000 (0.990, 1.011)
Male	1.081 (0.821, 1.423)	0.969 (0.725, 1.296)	1.109 (0.821, 1.501)
Non-Binary/Other	0.952 (0.202, 3.451)	1.279 (0.059, 13.499)	1.115 (0.048, 12.773)
Sexual Orientation: Other	1.095 (0.690, 1.701)	0.681 (0.344, 1.264)	0.685 (0.339, 1.300)
Higher Education			1.643 *** (1.215, 2.218)
Healthcare Worker: Yes			2.386 *** (1.663, 3.419)
Work with Animals: Yes			2.291 *** (1.328, 3.927)
Non-Heterosexual Males		2.969 ** (1.189, 7.584)	2.788 ** (1.091, 7.283)
Non-Binary/Other Non-Heterosexual		0.986 (0.049, 29.602)	1.646 (0.076, 52.886)
Intercept	0.427 *** (0.281, 0.646)	0.451 *** (0.296, 0.686)	0.228 *** (0.139, 0.369)
Observations	1055	1055	1055
Log Likelihood	−612.670	−609.893	−584.938
Akaike Inf. Crit.	1235.341	1233.786	1189.875

Note: ** *p* < 0.05; *** *p* < 0.01.

## Data Availability

All relevant data are within the paper. Dataset available on reasonable request from the corresponding author.

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
