# Peer review of "Assessment of Italian Population Awareness on One-Health, Zoonoses and the Mpox Vaccine: A Nationwide Cross-Sectional Study"

_vaccines, 2024, doi:10.3390/vaccines12030258_

Round 1

Reviewer 1 Report

Comments and Suggestions for Authors

This is a well-polished study there is no major concern associated with this manuscript. One minor comment: The monkeypox is renamed as mpox. Please revise accordingly throughout the manuscript.

Author Response

We would like to thank the reviewer for this comment. We revised “mpox” throughout the manuscript.

The manuscript attached contains all the revisions requested by the reviewers, which are highlighted in blue

Reviewer 2 Report

Comments and Suggestions for Authors

Review 09.02.2024

Vaccines  https://www.mdpi.com/journal/vaccines

Assessment of Italian Population Awareness on One-Health, Zoonoses and the Monkeypox Vaccine: a nationwide cross-sectional study - Bert, et al

Background: The article is overall a well written manuscript from a group of public health individuals from Turin Italy. They attempt to determine the “awareness’ of two important public health issues, Mpox (CDC/WHO-more acceptable term to monkey pox) and the vaccine and the One Health concept. The outbreak of this zoonotic infection caught the world by surprise perhaps related to our over lack of appreciation of global health issues and the realization of how susceptible we are due to changes in world interconnectivity. On reading it was somewhat predictable that the general Italian sampled cohort including health workers would not have a strong knowledge of Mpox or the vaccine. This certainly was the case in the US and for a clinician- researcher such as myself I had not heard of Mpox since my medical school days.

The findings somewhat bear this out although I have serious concerns on the survey and size sampling outlined below. Similar issues with understanding the One-Health program which is highly laudable but not well advertised globally and certainly in the US. This such an important program as illustrated by the recent COVID pandemic with direct implications of health connectivity.

The potentially most important aspect of this project is to understand how the general population and the health community in Italy obtains and trusts public health information such as related to new diseases and in particular zoonotic infection that are foreign to a country.

I will focus my review based on the format from the journal.

As always, my comments are to be taken as constructive, professional and with the highest respect for the authors and their efforts.

Does the introduction provide sufficient background and include all relevant references?

The introduction- background is well written and provides sufficient referenced material for a reader of Vaccines to understand the aims of the project. It appears all the relevant references were included, and I found no conflict of interest. I assume the authors did not use one of the AI platforms such as chat GPT. Its increasingly common we review our writing using one of these AI programs.

The authors provided an excellent description of primary prevention of zoonotic infections which was quite educational. 

The primary -secondary aim are well stated as shown-

“Therefore, the present study was designed with the primary aim of exploring the general population’s knowledge regarding One Health and Zoonoses, delving not only into their knowledge but also potential predictors and determinants, with the ultimate goal of identifying any subgroups that could hypothetically be targeted for future awareness and health education campaigns.

Secondarily, this study focused on assessing awareness of the existence of a vaccine against the Monkey Pox virus, a pathogen that spread significantly across various European countries, especially during the summer of 2022.”.

Based on these interesting aims the design utilized a survey submitted through various social media platforms that appear from my review of the literature utilized by over 80% of the Italian population.

Is the research design appropriate?

Are the methods adequately described?

The use of questionaries- surveys is at the heart of data gathering and understanding critical public health issues. Based on the aims the survey was constructed with 45 questions and separated into 4 parts The survey included a validated health literacy question which was an excellent inclusion. The 4 parts appear to be comprehensive and do address determinants that would fulfill the specific aims. I would request the actual questionnaire in both Italian and English translated be provided in a supplement or appendix.

I do have some concerns related to the design:

1.     How were the de-identified responses collected from the various social media platforms collected? Qualtrics, Survey monkey, RedCap etc? This should be described briefly in the material and methods.

2.     Did the Italian IRB verify all collected information including demographics were de-identified and as such could not be used to back engineer identification.

3.     The use of “question borrowing” from other studies using surveys and questionnaires is rather common but there are two issues- copyright protection and subsequent validation of a newly constructed “hybrid” survey. The authors should note that using questions from other published surveys was at least approved by the original primary investigators or copyright permission holders. The second issue is more problematic. It doesn’t appear that the 45-question survey was validated using accepted protocol. The obvious implications are the internal and external validity of this survey completed by a very small segment of the Italian population. The authors should address my concern and provide support for the validity of the survey.

4.     Sample size and generalizability- The authors addressed this important study limitation at the end of the article. But I would like to know more on how many potential respondents were in the population that did not access the survey. Provide a denominator to understand better the sampling validity. The authors also noted which I completely agree is this sampled cohort may not truly represent all the Italian population as it was distributed vis social media.  Increasing sampling by other techniques or strategies may of course increase resources and time for data collection but would have better validated the findings. As such I believe the results from this very small cohort may represent the highest level of Mpox, One-Health and vaccine knowledge and if generated to the whole population the findings would be even more dramatic.

Are the results clearly presented?

Overall, the results are presented well but the tables 4,5,7 are difficult to read and take some time to evaluate. It’s important data but perhaps some additional formatting and simplification would improve readability.

Specific comments on selected results:

3.1- The authors break the SOGI but do not provide a number in Italy of the total- hetero, homosexual other. Would be instructive to get a sense of the sampling.

A GAD scoring instrument was included. In the US-IRBs reviewing projects Identifying subjects with a high level of a mental health condition – generalized anxiety- in a survey requires some additional counseling or education pathway for the subject. For example, providing resources so the subject can contact an Italian mental health provider. The rationale is based on the possibility the GAD may uncover newly discovered anxiety but no follow-on instructions for additional care.

3.2- Key section because it identifies sources of information in general for this small cohort. It’s interesting that most of the information is not through social media, but the survey was provided through social media.

The findings will set the stage for the Italian public health authorities to redesign the way information is made available to the public related to rare conditions such as novel zoonotic diseases.

3.6- Regression modeling – very interesting and important section to “tease out” potential confounders variables in the outcomes. I studied the 3 models but the following statement is a bit obtuse.

“Comparing the first two models with Model 3 through a likelihood ratio test (LRT), the deviance of residuals was lower in the latter, both compared to Model 1 (p 0.001) and Model 2 (p 0.001).”

Many readers of Vaccines may not have the same level of statistical sophistication as the Italian authors so I would provide a very straightforward explanation of this test as it’s a critical issue with the modeling. 

Overall- A well written article with near perfect English attempting to discover very important public health issues related to an emerging health concern- how to provide accurate and information to the public, how does the public access that information and do they trust the sources of information. The other important aspect is how public health programs are being disseminated ONE-HEALTH and accepted by the public. It’s always difficult to educate the public as to the importance of these programs but the experience with the Mpox and vaccine could serve as a prototype to redesign education and distribution of these important issues. The reviewer does have some serious concerns that do need to be addressed as noted above. 

Author Response

We would like to thank the reviewer for the relevant issues that have been raised. We have addressed the comments as reported in the following paragraphs.

We are glad our Introduction was complete and clear.

The use of questionnaires- surveys is at the heart of data gathering and understanding critical public health issues. Based on the aims the survey was constructed with 45 questions and separated into 4 parts The survey included a validated health literacy question which was an excellent inclusion. The 4 parts appear to be comprehensive and do address determinants that would fulfill the specific aims. I would request the actual questionnaire in both Italian and English translated be provided in a supplement or appendix.

We provided the questionnaire in both Italian and English translated versions in Supplementary Materials.

How were the de-identified responses collected from the various social media platforms collected? Qualtrics, Survey monkey, RedCap etc? This should be described briefly in the material and methods.

We appreciate this suggestion. We added the following sentence in the Methods: “The questionnaire was set on the LimeSurvey platform, provided by the University of Turin. No information allowing the identification of the participant nor Internet Protocol (IP) addresses has been recorded.

Did the Italian IRB verify all collected information including demographics were de-identified and as such could not be used to back engineer identification.

The Ethical Committee of the University of Turin did not verify our dataset after the data collection, but it approved the protocol and the questionnaire. We guaranteed through Limersurvey that “No information allowing the identification of the participant nor Internet Protocol (IP) addresses has been recorded.” and “Prior to questionnaire administration, information regarding the purpose, objectives, and characteristics of the study was provided, and participants were asked for their informed consent by continuing with the questionnaire.”, as we have now clarified in our Methods section.

The use of “question borrowing” from other studies using surveys and questionnaires is rather common but there are two issues- copyright protection and subsequent validation of a newly constructed “hybrid” survey. The authors should note that using questions from other published surveys was at least approved by the original primary investigators or copyright permission holders. The second issue is more problematic. It doesn’t appear that the 45-question survey was validated using accepted protocol. The obvious implications are the internal and external validity of this survey completed by a very small segment of the Italian population. The authors should address my concern and provide support for the validity of the survey.

We understand the concern of the reviewer. The validated tools we used are not covered by a license. Instead, the other questions were not exactly copied from other papers, but we were inspired from these articles to understand which items to study and include in our questionnaire, citing these papers in the methods as references.

To provide data about the reliability of our main outcomes’ scores, we added the following information in the Methods section: “The Cronbach's alpha coefficient was calculated to assess the consistency of the questions in understanding the level of knowledge on zoonoses and One Health.” In the Results section: “The Cronbach's coefficient yielded a consistency score of 0.74 for One-Health domain.” “The consistency of the score on zoonoses, obtained through the Cronbach's coefficient, was 0.81.”. 

In addition, we added the following limitation in the Discussion: “In addition, it should be noted that, although these outcomes’ scores had good reliability, they were not validated.”

Sample size and generalizability- The authors addressed this important study limitation at the end of the article. But I would like to know more on how many potential respondents were in the population that did not access the survey. Provide a denominator to understand better the sampling validity. The authors also noted which I completely agree is this sampled cohort may not truly represent all the Italian population as it was distributed vis social media.  Increasing sampling by other techniques or strategies may of course increase resources and time for data collection but would have better validated the findings. As such I believe the results from this very small cohort may represent the highest level of Mpox, One-Health and vaccine knowledge and if generated to the whole population the findings would be even more dramatic.

Unfortunately, we do not have the denominator of potential respondents: we shared the questionnaire on social media; however, we cannot know in any way how many people the questionnaire has actually reached. Thus, we added the following limitation: “Also, it should be noted that through this opportunistic sampling method, we were unable to determine the total number of people reached by the questionnaire.”

Lastly, we agree with the reviewer: considering the potential pitfalls of the sampling, which are discussed in our limitations, the results of our study do underline the importance of planning interventions targeted  to the general population to address this issue.

 Overall, the results are presented well but the tables 4,5,7 are difficult to read and take some time to evaluate. It’s important data but perhaps some additional formatting and simplification would improve readability.

We used the formatting template provided by Vaccines. However, if allowed by the journal, we are willing to modify the formatting and aesthetics of the tables.

The authors break the SOGI but do not provide a number in Italy of the total- hetero, homosexual other. Would be instructive to get a sense of the sampling.

We appreciate this suggestion. We added the following information: “Considering representativeness, it should be noted that, in our sample, the percentage of individuals who self-identified as non-heterosexual was just slightly higher compared to the latest available data, which indicate that the LGBT+ population accounts for 9% in Italy [25].

A GAD scoring instrument was included. In the US-IRBs reviewing projects Identifying subjects with a high level of a mental health condition – generalized anxiety- in a survey requires some additional counseling or education pathway for the subject. For example, providing resources so the subject can contact an Italian mental health provider. The rationale is based on the possibility the GAD may uncover newly discovered anxiety but no follow-on instructions for additional care.

Our IRB did not raise this issue, thus we did not provide resources to our participants.

Key section because it identifies sources of information in general for this small cohort. It’s interesting that most of the information is not through social media, but the survey was provided through social media.

The findings will set the stage for the Italian public health authorities to redesign the way information is made available to the public related to rare conditions such as novel zoonotic diseases.

We agree with the reviewer. We hope that our results will inform future public health communication initiatives.

Regression modeling – very interesting and important section to “tease out” potential confounders variables in the outcomes. I studied the 3 models but the following statement is a bit obtuse.

“Comparing the first two models with Model 3 through a likelihood ratio test (LRT), the deviance of residuals was lower in the latter, both compared to Model 1 (p ≤ 0.001) and Model 2 (p ≤ 0.001).”

Many readers of Vaccines may not have the same level of statistical sophistication as the Italian authors so I would provide a very straightforward explanation of this test as it’s a critical issue with the modeling.

We clarified as follows in the Methods: “Regression models were compared using the Likelihood Ratio Test (LRT) a statistical test used to assess if the inclusion of additional parameters in a more complex model significantly improves its fit compared to a simpler model.”. Moreover, in results we clarified as follows: “When comparing the "reduced" models to Model 3 using a likelihood ratio test (LRT), the deviance of the residuals was lower in the latter, both compared to Model 1 (p ≤ 0.001) and Model 2 (p ≤ 0.001).This indicates that Model 3 captures a greater proportion of the data variability and offers a more comprehensive explanation for the observed outcomes compared to both Model 1 and Model 2.”

Overall- A well written article with near perfect English attempting to discover very important public health issues related to an emerging health concern- how to provide accurate and information to the public, how does the public access that information and do they trust the sources of information. The other important aspect is how public health programs are being disseminated ONE-HEALTH and accepted by the public. It’s always difficult to educate the public as to the importance of these programs but the experience with the Mpox and vaccine could serve as a prototype to redesign education and distribution of these important issues. The reviewer does have some serious concerns that do need to be addressed as noted above.

We would like to thank the reviewer for these interesting observations about our study. We hope that our revisions have improved the paper.

Reviewer 3 Report

Comments and Suggestions for Authors

Estimated Authors,

I've been invited to review the paper from Bert et al. on the assessment of Italian Population Awareness on One-Health, Zoonoses and the Monkeypox Vaccine. Through a nationwide cross-sectional study, Authors provide an accurate and well designed analysis of factors associated with a good understanding of One Health and Zoonosis in the Italian population. The eventual sample was >1,000 individuals: therefore, despite some inconsistencies with the general Italian population (i.e. demographics, education, etc) that in turn impair the overall representativity of the study population, it retains both significance and interest for potential readers.

From the point of view of the present reviewer, however, some shortcomings do affect the study, and would require a series of improvements in order to guarantee the eventual acceptance of the study.

First, please provide as an annex / supplementary material the full text of the questionnaire: according to the text of the materials and methods section, one health was ascertained through a broad and wholesome approach. As a consequence, Readers should be able to ascertain which items (and therefore which topics) have been actually assessed by Authors. 

Second, Authors should provide some insights about the internal consistency of their questionnaire (e.g. calculation of Cronbach's alpha?).

Third, it would be of some interest to share with the readers which items did mostly contribute to the summary score, which ones were more frequently associated with wrong answers, etc.

Regarding your eventual results, you've noticed that living in towns > 50,000 inhabitants was positively associated with outcomes from model 1 and model 3 on zoonoses, but not on one health. Could be this habit associated with factors such as baseline literacy, occupation, etc? For example, were professionals with a background in human medicine, with higher education, or with better economic status (that in turn could be associated with higher educational achievements) clustered in larger towns? 

Moreover, Authors should provide some further details about the recruitment of participants: how was the questionnaire shared? Which social media / discussion groups were initially involved? As internet-based questionnaires are usually associated with some degree of self-recruitment, Authors should provide as much as possible information about the initial stages of the study in order to ascertain whether the original composition of the study was or not biased by the recruitment strategy.

Finally, I must confess some doubts about the consistency between the first sections of the paper with the mpox analyses. In fact, not only the main section of the study was performed in a different timeframe (2023 vs. 2022), but mpox is a nowadays mostly associated with interhuman spreading (and WHO promoted the revision of its official name also for this reason). In other words, assessing the mpox within a One-Health questionnaire study could be somehow misleading. I would suggest Authors to remove mpox section from the main text.

Author Response

First, please provide as an annex / supplementary material the full text of the questionnaire: according to the text of the materials and methods section, one health was ascertained through a broad and wholesome approach. As a consequence, Readers should be able to ascertain which items (and therefore which topics) have been actually assessed by Authors. 

We provided the questionnaire in both Italian and English translated versions in Supplementary Materials. 

Second, Authors should provide some insights about the internal consistency of their questionnaire (e.g. calculation of Cronbach's alpha?).

We would like to thank the reviewer for this suggestion. To provide data about the reliability of our main outcomes’ scores, we added the following information in the Methods section: “The Cronbach's alpha coefficient was calculated to assess the consistency of the questions in understanding the level of knowledge on zoonoses and One Health.” In the Results section: “The Cronbach's coefficient yielded a consistency score of 0.74 for One-Health domain.” “The consistency of the score on zoonoses, obtained through the Cronbach's coefficient, was 0.81.”.

Third, it would be of some interest to share with the readers which items did mostly contribute to the summary score, which ones were more frequently associated with wrong answers, etc.

We agree with the reviewer. We added the following information: “Considering individual responses regarding the topic of zoonoses, only 373 participants, representing 35.2% of the total, identified Ebola as a zoonosis, and 294 participants, comprising 27.8% of the total, were aware of the prevalence of zoonoses compared to the total number of infectious diseases. A slightly higher percentage, 43.2%, equivalent to 457 out of 1058 participants, believes that some zoonoses are preventable through vaccination.” and “In addition, Evaluating individual responses on the topic of One Health, only 250 participants, representing 23.6% of the total, were aware that human activity impacts the emergence of new diseases, while fewer than 350 participants, representing about 30% of the total, were aware that One Health also addresses the psychological well-being of workers and secondary prevention.

Regarding your eventual results, you've noticed that living in towns > 50,000 inhabitants was positively associated with outcomes from model 1 and model 3 on zoonoses, but not on one health. Could be this habit associated with factors such as baseline literacy, occupation, etc? For example, were professionals with a background in human medicine, with higher education, or with better economic status (that in turn could be associated with higher educational achievements) clustered in larger towns? 

As described in the Methods, for each model, we checked for multicollinearity, meaning whether 2 or more covariates were strongly associated with each other. Additionally, adding the aforementioned variables into a single model seems to help reduce the deviance of residuals. It is therefore possible that living in a large municipality by itself contributes to an increase in awareness of these two issues, but understanding the reasons behind this phenomenon is challenging, and further studies would be needed to more comprehensively grasp.

Moreover, Authors should provide some further details about the recruitment of participants: how was the questionnaire shared? Which social media / discussion groups were initially involved? As internet-based questionnaires are usually associated with some degree of self-recruitment, Authors should provide as much as possible information about the initial stages of the study in order to ascertain whether the original composition of the study was or not biased by the recruitment strategy.

We clarified in the Methods as follows: “The sampling method employed was opportunistic. The questionnaire has been posted on neutral social groups, not focusing on one health, zoonoses, vaccines or related themes to avoid selection bias.” Unfortunately, we do not have the denominator of potential respondents: we shared the questionnaire on social media; however, we cannot know in any way how many people the questionnaire has actually reached. Thus, we added the following limitation: “Also, it should be noted that through this opportunistic sampling method, we were unable to determine the total number of people reached by the questionnaire.”

Finally, I must confess some doubts about the consistency between the first sections of the paper with the mpox analyses. In fact, not only the main section of the study was performed in a different timeframe (2023 vs. 2022), but mpox is a nowadays mostly associated with interhuman spreading (and WHO promoted the revision of its official name also for this reason). In other words, assessing the mpox within a One-Health questionnaire study could be somehow misleading. I would suggest Authors to remove mpox section from the main text.

Thank you for your feedback. While we acknowledge your doubts, we believe it's important to retain this section as it aligns with our secondary aim, which was not primarily focused on One Health and zoonoses but on vaccination attitudes.

Round 2

Reviewer 2 Report

Comments and Suggestions for Authors

Thank you for your thoughtful comments which address all of my concerns

Reviewer 3 Report

Comments and Suggestions for Authors

Estimated Authors,

the paper has been amended according to my previous report.

Even though I'm still sharing some doubts about the consistence of the study about zoonoses with the mpox survey (a disorder with inter-human spreading and a substantial share of cases among non-binary individuals, as otherwise acknowledged across the main text), the reply I've received are appropriate. In other words, I've no further requests and I'm endorsing the acceptance of the paper.